# RETRACTED: *Rosa davurica* Pall. Improves *Propionibacterium acnes*-Induced Inflammatory Responses in Mouse Ear Edema Model and Suppresses Pro-Inflammatory Chemokine Production via MAPK and NF-κB Pathways in HaCaT Cells

**DOI:** 10.3390/ijms21051717

**Published:** 2020-03-03

**Authors:** Du Hyeon Hwang, Dong Yeol Lee, Phil-Ok Koh, Hye Ryeon Yang, Changkeun Kang, Euikyung Kim

**Affiliations:** 1Institute of Animal Medicine, Gyeongsang National University, Jinju 52828, Korea; pooh9922@hanmail.net (D.H.H.); 2015210922@gnu.ac.kr (H.R.Y.); ckkang@gnu.ac.kr (C.K.); 2Anti-Aging Research Group, Gyeongnam Oriental Anti-Aging Institute, Sancheong 52230, Korea; dylee1984@gnoai.or.kr; 3Department of Anatomy, College of Veterinary Medicine, Gyeongsang National University, Jinju 52828, Korea; pokoh@gnu.ac.kr; 4Department of Pharmacology and Toxicology, College of Veterinary Medicine, Gyeongsang National University, Jinju 52828, Korea

**Keywords:** *Rosa davurica* Pall., *Propionibacterium acnes*, anti-inflammation, cytokines, MAPK, NF-κB, ear swelling

## Abstract

Acne, also known as acne vulgaris, is a common disorder of human skin involving the sebaceous gland and *Propionibacterium acnes* (*P. acnes*). Although there are a number of treatments suggested for acne, many of them have limitations in their safety and have efficacy issues. Therefore, there is a high demand to develop safe and effective novel acne treatments. In the present study, we demonstrate the protective effects of *Rosa davurica* Pall. leaves (RDL) extract against *P. acnes*-induced inflammatory responses in vitro and in vivo. The results showed that RDL dose-dependently inhibited the growth of skin bacteria, including *P. acnes* (KCTC3314) and aerobic *Staphylococcus aureus* (KCTC1621) or *Staphylococcus epidermidis* (KCTC1917). The downregulation of proinflammatory cytokines by RDL appears to be mediated by blocking the phosphorylations of mitogen-activated protein kinase (MAPK) and subsequent nuclear factor-kappa B (NF-κB) pathways in *P. acnes*-stimulated HaCaT cells. In a mouse model of acne vulgaris, histopathological changes were examined in the *P. acnes*-induced mouse ear edema. The concomitant intradermal injection of RDL resulted in the reduction of ear swelling in mice along with microabscess but exerted no cytotoxic effects for skin cells. Instrumental analysis demonstrated there were seven major components in the RDL extract, and they seemed to have important roles in the anti-inflammatory and antimicrobial effects of RDL. Conclusively, our present work showed for the first time that RDL has anti-inflammatory and antimicrobial effects against *P. acnes*, suggesting RDL as a promising novel strategy for the treatment of acne, including natural additives in anti-acne cosmetics or pharmaceutical products.

## 1. Introduction

Acne vulgaris is one of the most common inflammatory skin disorders, which are caused by bacterial infections, including Gram-positive (*Propionibacterium acnes* and *Staphylococcus epidermidis)* as well as Gram-negative (*Pseudomonas aeruginosa*) bacteria [1,2]. *Propionibacterium acnes* (*P. acnes*) is a ubiquitous member of the skin microbiota and is frequently found in sebaceous follicles located on the face, chest and the back of the majority of humans [3]; whereas, *Staphylococcus aureus* and *Staphylococcus epidermidis* are members of normal human skin flora, which commonly cause skin abscesses, ulcers and soft tissue infections [4,5]; they can also aggravate the acne inflammatory event but less significantly than *P. acnes* [5,6]. Acne is a multifactorial inflammatory disease, and there are four processes that are considered to have a pivotal role in the formation of acne lesions, namely, increased sebum production, altered follicular keratinization, inflammation and bacterial colonization of the pilosebaceous unit by *P. acnes* [7,8].

Currently, various therapeutic agents involving steroids and/or antibiotics have been proposed for acne, to inhibit inflammation or kill bacteria [6]. However, these therapeutic agents may lead to the emergence of resistant pathogens and side effects [9,10]. Treatment of acne vulgaris with commonly used antibiotics, including oral tetracycline and topical erythromycin and clindamycin, has increased the resistance of *P. acnes* to those antibiotics and thereby increased the likelihood of therapeutic failure [11]; hence, the development of more efficacious and safe drugs has been strongly demanded.

Recently, the public interest in acne has been rapidly augmented, and some natural compounds have been presented as an alternative treatment in acne therapy. Among them, rosemary and wild bitter melon, and their bioactive constituents such as rosmarinic acid, phytol, lutein and phenolic compounds have been demonstrated to suppress *P. acnes*-induced proinflammatory cytokine releases [12,13,14].

*Rosa davurica* Pall. (Rosaceae) is a deciduous shrub, mainly distributed in the northeastern parts of Asia. *R. davurica* has been applied as a traditional Chinese herbal therapy for various diseases, and the medicinally beneficial effects of *R. davurica*, such as their antioxidant activities, anti-HIV and antiviral, antibiotic and hypoglycemic activities are already well understood [12,13,15,16,17]. Moreover, contemporary research has shown that *R. davurica* possesses significant anti-inflammatory effects. Recently, It has been demonstrated that the leaves of *R. davurica* possess antiangiogenic as well as related anti-inflammatory and antinociceptive activities [15]. However, the anti-inflammatory and related activity mechanisms of *R. davurica* against acne have yet to be revealed. In this study, we demonstrated the anti-inflammatory effects of *Propionibacterium* acnes of *R. davurica* Pall. leaves (RDL) in vitro and in vivo. The results of this study provide information on the development of an alternative acne therapeutic agent.

## 2. Result

### 2.1. Antimicrobial Effects of RDL against Skin Bacteria

The antimicrobial effect of various parts of *Rosa davurica* Pall. extract was investigated against skin microorganisms, namely *P. acnes*, *Staphylococcus aureus* (*S. aureus*) and *Staphylococcus epidermidis* (*S. epidermidis*) by agar diffusion assay. As shown in Figure 1, all tested skin bacteria were highly sensitive to the different parts of *Rosa davurica* Pall. extracts. In addition, subsequent experiments were conducted to determine the inhibitory concentrations of various parts of *Rosa davurica* Pall. extract. *Rosa davurica* Pall. extract showed a potent antimicrobial effect. The lowest minimum inhibitory concentration (MICs) against *P. acnes* (62.5 μg/mL), *S. epidermidis* (125 μg/mL) and *S. aureus* (62.5 μg/mL) were recorded in RDL (Table 1). Similarly lowest minimum bactericidal concentration (MBCs) against *P. acnes* (125 μg/mL), *S. epidermidis* (125 μg/mL) and *S. aureus* (250 μg/mL) were also observed in RDL. Among all the extracts, the RDL exhibited a significantly greater inhibitory effect against skin bacteria than any other parts of the plant (*p* < 0.05) (Figure 1B; Table 1).

### 2.2. RDL Suppresses P. Acnes-Induced Expression of Proinflammatory Cytokines and TLR in HaCaT Cells

First of all, the cytotoxicity of RDL on HaCaT cell was determined before evaluating its anti-inflammatory activity. To do this, HaCaT cell was incubated with various concentrations of RDL for 24, 48 and 72 h. RDL showed no toxicity on the cells at all the concentrations tested in the present study (Figure 2); on the contrary, it has been increased in a time- and concentration-dependent manner. Based on these results, we selected 10 to 100 μg /mL RDL for 24 h, which appeared to be concentrations which had no effect.

In order to assess the effects of RDL on *P. acnes*-associated inflammatory changes, ELISA (Figure 3A) and Western blot (Figure 3B,C) analyses were employed to analyze the expression of proinflammatory cytokines. As shown in Figure 3A, upon treatment of *P. acnes*, the secretions of TNF-α, IL-8 and IL-1β significantly increased in HaCaT cells, while treatment with RDL significantly suppressed the *P. acnes*-induced TNF-α, IL-8 and IL-1β production. To further investigate the effect of RDL on *P. acnes*-induction of cytokines, protein expressions of these cytokines were evaluated by Western blot analysis. As shown in Figure 3B, *P. acnes* strongly augmented the expressions of TNF-α, IL-8 and IL-1β, whereas RDL treatment dramatically suppressed the induction of those gene expressions in *P. acnes*-treated cells. Since TLR activation may lead to the production of inflammatory cytokines, further Western blot analysis was performed on HaCaT cells. As observed in Figure 3C, RDL dose-dependently suppressed *P. acnes*-induced TLR2 expression. These results indicate that RDL specifically inhibits the expression of inflammatory cytokines and TLR induced by *P. acnes* in HaCaT cells.

### 2.3. Modulation of P. Acnes-Induced Inflammatory Cell Signaling by RDL

We evaluated the molecular mechanism of *P. acnes*-induced inflammation and its cell signaling, such as NF-κB and MAPK, including extracellular signal-related kinase (ERK) and p38 mitogen-activated kinase (p38) in HaCaT cells. Figure 4A shows that the levels of phosphorylated p38 and ERK were significantly increased in *P. acnes*-exposed cells. RDL treatment dose-dependently inhibited the phosphorylation of p38 and ERK triggered by *P. acnes*. NF-κB activation often leads to the expressions of proinflammatory cytokines and transcriptional regulator of multiple genes. As shown in Figure 4B, RDL treatment impeded the phosphorylation of IκBα and translocation of NF-κB in a dose-dependent manner. These findings suggest that RDL might have a therapeutic role in acne due to its inhibitory effect on inflammatory cell signaling, via the downregulation of MAPK and NF-κB pathways.

### 2.4. Effect of RDL on P. Acnes-Induced Inflammation In Vivo

To examine the antimicrobial effect of RDL in vivo, ICR mouse ears were intradermally injected with *P. acnes*. Ears of mice were injected intradermally with 6 × 10^7^ CFU/20 μl in PBS of *P. acnes* (left ears). RDL (100, 300 and 1000 μg/20 μL in PBS) was applied to the surface of the skin of the right ear of mice in each group after the injection of *P. acnes* (Figure 5). Significant ear swelling could be observed from *P. acnes*-injected ears 24 h after the injection. Also, histological observation of the injected tissue revealed that *P. acnes* triggered a considerable increase in the infiltration of inflammatory cells to the site. In contrast, no swelling was observed from the PBS alone-injected ear (vehicle control; Figure 5A). Injection of RDL significantly modulated *P. acnes*-induced ear swelling (Figure 5B,C) and the number of *P. acnes* colonized within the ear (Figure 5D). Histological images also indicated that RDL treatment reduced the development of granulomatous changes in response to *P. acnes* exposure compared with the injection of an equal amount of PBS (Figure 5A). To examine the reduction in ear inflammation by RDL treatment, we next measured inflammatory cytokine levels, using ELISA (Figure 5E). These inflammatory cytokines were consistently decreased by RDL treatment. Collectively, these results suggest that RDL improved *P. acnes*-induced skin inflammation in the mouse model.

### 2.5. Measurement of the Representative Component in RDL by GS-MS Analysis

We investigated which constituent of RDL is involved to usefully treat *P. acnes*-induced inflammatory pathways. The molecular components present in the extract of RDL was determined by GC-MS analysis (Figure 6). A total of seven compounds were identified in the RDL extract are shown in Table 2, along with their structure and reported activities. The major constituents present in the extract were 4H-pyran-4-one, 2,3-dihydro-3,5-dihydroxy-6-methyl-, (1); 1,2,3-bezenetriol, (2); n-hexadecanoic acid, (3); 1-hexadecanol, (4); 9,12-octadecadienoic acid (Z, Z), (5); 9-octadecadienoic acid, (E), (6) and octadecadienoic acid, (7). They are over 90% matched by the MS library in the NIST AMDIS version 2.1 software. Previously, according to Dr. Duke’s database, these seven main ingredients have been reported to show antimicrobial and anti-inflammatory effects.

## 3. Discussion

Acne vulgaris is one of the most common chronic dermatological diseases, affecting 80–85% of teenagers globally [18,19,20]. *P. acnes*, as a major causing factor, can colonize the pilosebaceous follicle, resulting in the inflammatory reaction in acne vulgaris. In infected sebaceous follicles, *P. acnes* releases lipases, proteases and hyaluronidases, which are suspected to damage skin tissues in acne vulgaris patients [21]. *P. acnes* can also induce inflammatory reactions with keratinocytes, sebaceous gland cells and monocytes, indicating its possible influence on the growth and differentiation of cells [22]. Although the current treatment options are effective, they are largely associated with limitations, such as unwanted side effects, chronicity, relapses and recurrences; hence there is strong demand to develop a novel therapeutic reagent for acne vulgaris. Accordingly, we evaluated the antibacterial effect of RDL extracts on *P. acnes* through disc diffusion assays. As a result, we showed for the first time that RDL inhibits more strongly the growth of skin bacteria, including anaerobic *P. acnes* (KCTC3314), aerobic *S. aureus* (KCTC1621) and *S. epidermidis* (KCTC1917), than does triclosan, an agent frequently used to treat acne. Although the antioxidant and anti-inflammatory activities of *Rosa davurica* Pall. and its components have been reported, little is known about their effects on *P. acnes*-induced inflammatory responses.

We provide here a preliminary description of the molecular basis of the anti-inflammatory action of RDL in *P. acnes*-stimulated HaCaT cells. It has been previously suggested that *P. acnes* may stimulate the development of inflammatory acne lesions by releasing chemotactic substances via the activation of toll-like receptors (TLR), which can cause the release of proinflammatory cytokines, including tumor necrosis factor (TNF)-α, interleukin (IL)-6, IL-8, IL-1β and IL-12 [23,24,25]. The TLR protein family has been identified as *P. acnes* responsive receptors, and the expression of TLR2 is elevated in *P. acnes*-infected keratinocytes, monocytes and macrophages [26]. Upon the stimulation of TLR signaling pathways, the innate immune system is able to recognize the microbial components and then induce cytokine/chemokine secretion in acne [23]. Several previous studies have shown that activation of these pattern TLR receptors induces the secretion of proinflammatory cytokines, such as TNF-α and IL-8, and exacerbates skin inflammation in mice [27]. Here, we have investigated if RDL can inhibit proinflammatory cytokine production in *P. acnes*-stimulated HaCaT cells. Upon the exposure of *P. acnes*, the secretions of IL-8, IL-1β and TNF-α have significantly increased in HaCaT cells; whereas, the cotreatment with RDL significantly suppressed the *P acnes* induced IL-8, IL-1β and TNF-α production. Also, NF-κB and mitogen-activated protein kinase (MAPK) pathways have been proposed by other investigators to be related to *P. acnes*-induced inflammatory cytokine production [28,29]. The phosphorylations of NF-κB and MAPKs can convey the signal to the nucleus, which then controls the expression of various inflammatory cytokines [30,31,32,33,34,35]. Therefore, the inhibition of MAPKs and NF-κB activation has been suggested as a promising therapeutic target for the anti-inflammatory strategy in *P. acnes*-induced inflammation. As shown in our data, treatment with RDL suppressed MAPK phosphorylation and NF-κB activation. The downregulation of proinflammatory cytokines by RDL may partially be mediated by blocking MAPK pathways and subsequent NF-κB activation. 

In order to investigate the therapeutic effects of RDL in vivo, we have used ICR mouse, a well-known animal model of acne vulgaris, in which the pattern of *P. acnes*-induced inflammation is similar to that of human acne lesions [36]. Briefly, the injection of *P. acnes* significantly induced ear swelling, redness and erythema in ICR mice and considerably increased the number of infiltrated inflammatory cells in the skin lesions [37]. We have also evaluated the effects of RDL on ear skin severity and histopathological changes in *P. acnes*-induced mouse ear edema model. Intradermal injection of RDL dose-dependently reduced ear edema symptoms, the thickness, weights and inflammatory cytokines compared with the *P. acnes*-challenged group without RDL treatment. In addition, the administration of RDL reduced the number of *P. acnes* colonized within the ear. The results suggest a plausible therapeutic effect of RDL associated acne model. In this study, we have investigated a plausible mechanism underlying the anti-inflammatory activity of RDL in HaCaT cells and also demonstrated its therapeutic potential for acne using *P. acnes*-induced mouse ear edema model. We found that RDL can suppress the expression of proinflammatory cytokines via MAPKs and NF-κB pathways, which are well known to be stimulated by *P. acnes* in HaCaT cells (Figure 7). 

Many previous reports have suggested that *Rosa davurica* Pall. contains a rich amount of phytochemicals, which may contribute to its various pharmacological effects [15,16,17]. The fruit and leaf of *Rosa davurica* Pall. are rich in vitamin C, and used in nutraceutical drinks [38]. The methanol extract of the plant roots has high levels of antioxidants and free radical scavenging activities. The extract appears to include phenolics, which may contribute to the antioxidant activity [16]. However, investigation of the inhibitory activity and mechanism of RDL extract on anti-inflammatory effects against *P. acnes* has not been reported. In the present study, a total of seven compounds identified in the RDL extract are shown. It can be concluded that, RDL contains various bioactive compounds; these compounds not only possess potent anti-inflammatory activity but also antimicrobial effects against *P. acnes*.

## 4. Materials and Methods

### 4.1. Chemicals and Reagents

Reinforced Clostridium Medium (RCM; BD Diagnostics, Sparks Glencoe, MD, USA), Tryptic soy broth (TSB; Soybean–Casein Digest Medium) and tryptic soy agar (TSA) were purchased from Difco™ (Thailand). FGM^TM^-2 BulletKit^TM^ (Lonza Group Ltd., Basel, Switzerland), Bovine serum albumin (BSA), Fetal bovine serum (FBS), penicillin, streptomycin and trypsin were purchased from Gibco-BRL (Grand Island, NY, USA). Dimethyl sulfoxide (DMSO), 3-(4,5-dimethylthiazol-2-yl)-2,5 diphenyltetrazolium bromide (MTT), 2-(4-iodophenyl)-3-(4-nitrophenyl)-5-phenyltetrazolium chloride (INT), Clindamycin and Triclosan were from Sigma-Aldrich Inc. (St. Louis, MO, USA). ELISA kits for TNF-α, IL-8, and IL-1β were obtained from BioLegend (San Diego, CA, USA). Antibodies for TNF-α, IL-8, IL-1β, MAPK, phosphor-MAPK, phosphor-IκBα, NF-κB and GAPDH were obtained from Cell Signaling Technology (Beverly, MA, USA). All other reagents used were of the purest grade available.

### 4.2. Plant Materials and Extract Preparation

The dried leaves of *Rosa davurica* Pall. were obtained from a local farm of Jungsun, Gangwon-do, South Korea. The samples were gently rinsed three times with fresh distilled water and dried at room temperature. The samples were lyophilized and homogenized with a grinder, then the powder was stored at −20 ˚C until use. For extraction, an aliquot (20 g) of lyophilized RDL powder was added to 400 mL of 70% methanol and shaken (150 rpm) for 24 h at room temperature, after which the extract was filtered through the Advantech No.3 filter paper (Osaka, Japan) and the methanol was evaporated using a rotary vacuum evaporator (Tokyo Rikakikai Co., Ltd., Tokyo, Japan). The final aqueous part was lyophilized and kept at −20 ˚C until use. The lyophilized powder was freshly dissolved in DMSO before the experiments.

### 4.3. Preparation of Bacteria

*P. acnes* (KCTC3314) was obtained from the Korean collection for type cultures (KCTC, Daejeon, South Korea) and cultured on RCM at 37 ˚C under anaerobic conditions using Gas-Pak (BD, Sparks, MD, USA) until it reached OD600 = 1.0 (stationary phase). *S. aureus* (KCTC1621) or *S. epidermidis* (KCTC1917) was cultured on TSA overnight at 37 ˚C. The bacteria from single colonies were cultured in TSB overnight at 37 ˚C. The overnight culture was diluted 1:20 and cultured again until the absorbance of the culture broth reaching around OD600 = 1.0. The bacteria were harvested by centrifugation at 5000 *g* for 10 min, washed with PBS and suspended to an appropriate amount of PBS for further experiments.

### 4.4. Antimicrobial Activity

#### 4.4.1. Disc Diffusion Method

Agar diffusion assay was performed to determine the antimicrobial activities of the extracts as described by Clinical and Laboratory Standards Institute (CLSI) with some modifications [6,39]. Briefly, *P. acnes* suspension was diluted with RCM to a final concentration of approximately 1 × 10^6^ CFU/mL (Optical density; OD600 = 0.1). The diluted *P. acnes* suspension (200 μL) was streaked onto reinforced clostridium agar. In addition, *S. aureus* and *S. epidermidis* suspensions were also diluted with TSB resulting in a final concentration of approximately 1 × 10^6^ CFU/mL (OD600 = 0.1), which were then streaked onto TSA. The agar was excavated and each sample, dissolved in 1% v/v DMSO in culture broth at various concentrations (0.1–10 mg/mL) of RDL methanol extract and was added into each well (6 mm diameter). The inoculated plates of *P. acnes* were incubated at 37 ˚C, under anaerobic conditions for 72 h, while those of *S. aureus* and *S. epidermidis* were incubated at 37 ˚C, for 24 h. The diameter of the inhibition zone was measured in millimeters. Triclosan (100 ppm) were used as positive controls. The culture broth served as a negative control, while 1% v/v DMSO in culture broth was considered as a solvent control. The experiments were performed in triplicate.

#### 4.4.2. Determination of Minimum Inhibitory and Bactericidal Concentrations

The minimal inhibitory concentration (MIC) and minimum bactericidal concentration (MBC) values were determined by the broth microdilution method with slight modifications [40,41]. Extracts of *Rosa davurica* Pall. were dissolved in DMSO (10% of total volume) and two-fold dilutions were done using presterilized culture broth to give final concentrations ranging from 1000–15.6 μg/mL, then 100 μL of each dilution was distributed in 96-well plates, as well as a sterility control (sterilized nutrient broth), growth control (culture broth with DMSO), and clindamycin was used as positive control. Each test and growth control well was inoculated with 5 μL of a bacterial suspension (10^8^ CFU/mL or 10^5^ CFU/well). All experiments were performed in triplicate, and the microdilution plates were incubated under optimum conditions. Bacterial growth was detected first by optical density determination and later by the addition of 20 μL of 70% alcoholic solution of INT (0.5 mg/mL) into each well, followed by incubation for 30 minutes. Where bacterial growth occurred, INT changed from yellow to purple. Before the addition of INT, a subculture was made from each well without apparent growth to determine MBC. The MBCs of the extracts were plated 10 μl of samples from each MIC well without visible growth onto culture media plates. Following the incubation for the optimum period, the plates were examined for colony growth and MBCs were recorded. MIC and MBC values were defined as the lowest concentration of each RDL, which completely inhibited growth or yielded no viable microorganisms, respectively.

### 4.5. Determination of the Anti-Inflammatory Activity

#### 4.5.1. Cell Culture

Human keratinocyte HaCaT cells were maintained in DMEM with 10% FBS and 100 units antibiotics. Cells were cultured at 37 °C in a humidified incubator under 5% CO_2_ atmosphere. HaCaT cell was kindly supplied by Dr. T-J Yoon (Gyeongsang National University, Jinju, Korea).

#### 4.5.2. MTT Assay for Cell Viability

Cell viability was measured by MTT (3-(4,5-dimethyl-2-yl)-2,5-diphenyltetrazolium bromide) reduction assay, as described previously [42] with a slight modification. Briefly, the cells were plated at a density of 1 × 10^5^ cells/well in 24 well plates and cultured overnight in complete DMEM media containing 10% FBS. The cells were then treated with various concentrations (0–300 μg/mL) of RDL and incubated for 24, 48 and 72 hr in serum-free medium. MTT dye (5 mg/mL) was added to the cell culture media and they were incubated for an additional 3 hr. After the medium was removed, DMSO was added to the cells for the solubilization of generated formazan salts. The amount of formazan salt formation was decided by measuring the optical density (OD) at 540 nm using GENios® microplate spectrophotometer (PowerWaveTMXS, BioTek Instruments, Inc., Winooski, VT, USA). Relative cell viability of treatment was calculated as a percentage of vehicle-treated control; OD of treated cells/OD of control × 100.

#### 4.5.3. Cytokine Measurements Using ELISA Assay

The anti-inflammatory activity of RDL was examined in *P. acnes*-stimulated HaCaT cells. To prepare, the *P. acnes* culture was harvested and washed with PBS and then centrifuged at 10,000× *g* for 5 min. After two washes with PBS, the *P. acnes* pellet was resuspended in DMEM medium without antibiotics. After 30 min, the cells were treated with live *P. acnes* (wet weight 200 μg/mL) alone or in combination with various concentrations of RDL for 24 h incubation period. Then, cell-free supernatants were collected and the concentrations of TNF-α, IL-8 and IL-1β were determined using a commercially available ELISA kit, according to the manufacturer’s instructions. Reading of the absorbance at 450 nm was performed by GENios® microplate spectrophotometer (PowerWaveTMXS, BioTek Instruments, Inc., Winooski, VT, USA).

### 4.6. P. Acnes-Induced Inflammation In Vivo 

Eight-weeks-old male ICR mice were purchased from Samtako Inc. (Osan, Korea) and cared for in the Gyeongsang national university laboratory animal research center. The animal room was maintained at 23 ± 2 °C with a 12-h light/dark cycle. Food and tap water were supplied ad libitum. The animal study protocol used in this work was approved (July 30, 2018) by the Institutional Animal Care and Use Committee of Gyeongsang National University, and the animal study protocol number was GNU-180730-M0040.

*P. acnes* (6 × 10^7^ CFU per 20 μL in PBS) was intradermally injected into the left ear of ICR mice. The right ear of the same mice was injected with 20 μl of PBS (*n* = 5). Subsequently, the *P. acnes* and PBS-injected sites were intradermally injected with RDL (100, 300 and 1000 μg per 20 μL in PBS). As a control, an equal volume (20 μL) in PBS was injected into both ears.

At the end of each treatment period (24 h later), the animals were sacrificed and the ears were excised. Ear thickness and weight were measured on the day of sacrifice. The extent of edema was measured by the weight difference between the left and the right ear disk. The ear thickness and weight of *P. acnes*-injected ear were calculated as a percentage of PBS-injected control, respectively.

To determine *P. acnes* number in the ear, the ear was cut off and punched with a 6 mm biopsy punch 24 h after *P. acnes* injection. The punched biopsy sample was homogenized in 200 μL of sterile PBS with a hand tissue grinder. CFUs of *P. acnes* in the ear were enumerated by plating serial dilutions (1:10^2^ to 1:10^8^) of the homogenate on an RCM agar plate. To count colonies, the plate was anaerobically incubated for 72 h at 37 ˚C.

### 4.7. Western Blot Analysis

Western blotting assay was performed as previously described [43]. Briefly, the treated cells were rinsed twice with ice-cold PBS, and then added to 300 μL of RIPA (Radioimmunoprecipitation assay) buffer containing protease inhibitor cocktail. The sample proteins (20 μg) were run on 12% SDS-polyacrylamide gel electrophoresis, transferred to PVDF membranes (Bio-Rad, CA, USA) and subsequently subjected to immunoblot analysis using specific primary antibodies for overnight at 4 °C. The membranes were gently rinsed with wash buffer and incubated with horseradish peroxidase-conjugated secondary antibody (Cell Signaling Technology, Beverly, MA, USA) for 1 h at room temperature. The blots were visualized by using the enhanced chemiluminescence method (ECL; Amersham Biosciences, Buckinghamshire, UK) and analyzed using Chemi doc XRS (Bio-Rad, CA, USA). Densitometry analysis was performed with a Hewlett-Packard scanner and NIH Image software (Image J, National Institutes of Health, Bethesda, MD, USA).

### 4.8. Histological Analysis

The ear tissue slices were fixed in 10% neutralized formalin solution for at least 24 h at the room temperature. After the fixation, the tissues were embedded in paraffin wax, sectioned (5 μm), deparaffinized, and rehydrated, then stained with hematoxylin and eosin (H&E). Histopathological alterations were examined by light microscopy, and an arbitrary scope was given to each microscopic field viewed at a magnification of 200.

### 4.9. GC-MS Analysis

The GC-MS analysis was performed as previously described [44]. The RDL extract was analyzed by the comparison with Kovats gas chromatographic retention index (KI) [45] and by the mass spectrometry (MS) fragmentation pattern of each component compared with those of authentic chemicals. The concentration of each chemical was analyzed using a previously reported method [46]. An Agilent model 6890 GC, interfaced to an Agilent 5971A mass selective detector (GC/MS) was used for mass spectral identification of the GC components at MS ionization voltage of 70 eV. GC column conditions were exactly the same as the ones used for GC/FID.

Interpretation of mass spectrum GC-MS was performed using the database of National Institute Standard and Technology (NIST) having more than 62,000 patterns. The spectrum of the unknown component was compared to the known components stored in the NIST librarys.

### 4.10. Statistical Analysis

The results were expressed as mean ± standard deviation (S.D.). A paired student’s *t*-test was used to assess the significance of differences between two mean values. * *p* <0.05 and ** *p* <0.01 were considered to be statistically significant.

## 5. Conclusions

Acne, caused by gram positive bacteria is one of the most common inflammatory skin disorders. Though, anti-inflammatory-linked molecular mechanism for *R. davurica* against acne is still yet to be reported. The present results provide evidence that RDL reduces *P. acnes*-induced inflammation in the mouse ear edema model, by suppressing proinflammatory cytokines. Additionally, RDL exhibits significant antibacterial activities against skin microorganisms, including *P. acnes*. These findings of our study can support an essential knowledge on the evolution of new acne therapeutic agent for the treatment of acne vulgaris. Taken together, these suggest that RDL could potentially serve as an effective novel therapeutic agent for the treatment of acne vulgaris.

## Figures and Tables

**Figure 1 ijms-21-01717-f001:** Antimicrobial activity of of *Rosa davurica* Pall. against skin bacteria. (**A**) Effects of different parts of *Rosa davurica* Pall. (20 mg/mL) against skin bacteria. (**B**) Concentration-dependent inhibitory effects of *Rosa davurica* Pall. leaves (RDL) against skin bacteria. Skin bacteria tested were *P. acnes, S. epidermidis* and *S. aureus.* Triclosan was used as a positive control for the experiments. The diameter of the zone of inhibition was measured in millimeters. The experiments were performed in triplicate and the mean of the diameter of the inhibition zones was calculated. Statistically significant difference from vehicle dimethyl sulfoxide (DMSO) treatment, * *p* < 0.05 and ** *p* < 0.01.

**Figure 2 ijms-21-01717-f002:** The effect of various concentrations of RDL on the viability of human keratinocyte HaCaT cells. Exponentially growing HaCaT cells were plated onto 24-well plate and treated with indicated various concentrations (0–300 μg/mL) of RDL for 24, 48 and 72 h. The viability was measured by MTT assay as described in the materials and methods. The results were expressed as the mean ± SD of three experiments. Significant difference from control group, * *p* < 0.05 and ** *p* < 0.01.

**Figure 3 ijms-21-01717-f003:** RDL suppressed the proinflammatory cytokines and TLR2 in HaCaT cells. HaCaT cells were incubated for 24 h without *P. acnes* and with *P. acnes* alone, or with *P. acnes* in the presence of RDL. (**A**) ELISA results demonstrated that RDL suppressed the secretion of TNF-α, IL-8 and IL-1β in culture medium with HaCaT cells. (**B,C**) Western blot analysis showed that RDL inhibited the expression of TNF-α, IL-8, IL-1β and the regulation of TLR2. Data were expressed as the mean ± SD three independent experiments. * *p* < 0.05 and ** *p* < 0.01 compared to the *P. acnes* only treatment.

**Figure 4 ijms-21-01717-f004:** RDL suppresses the MAPK and NF-κB signaling pathway in *P. acnes*–treated HaCaT cells. HaCaT cells were incubated for 24 h without *P. acnes* and with *P. acnes* alone, or with *P. acnes* in the presence of RDL. (**A**) The effects of RDL on MAPKs, such as P38 and ERK phosphorylation, in HaCaT cells. P38 and ERK were quantified with ImageJ. (**B**) Effects of RDL on phosphorylation of IkB and NF-κB after *P. acnes* treatment of HaCaT cells. Western Blots were quantified using densitometry and normalized to total protein and GAPDH. p-IκB, NF-κB and p-NF-κB were quantified with ImageJ. Data were expressed as the mean ± SD three independent experiments. * *p* <0.05 and ** *p* <0.01 compared to the *P. acnes* only treatment.

**Figure 5 ijms-21-01717-f005:** The inhibitory effect of RDL *P. acnes*-induced inflammation. Ears of mice were injected intradermally with 6 × 10^7^ CFU/20 μl in PBS of *P. acnes* (left ears). RDL (100, 300 and 1000 μg per 20 μL in PBS) was applied to the surface of the skin of the right ear of mice in each group after the injection of *P. acnes*. (**A**) An increase in ear thickness and in the number of infiltrating inflammatory cells surrounding the injection site of *P. acnes* was observed by hematoxylin and eosin (H&E) staining; however, this increase was reversed in the presence of RDL. Scale bars represent 200 μm. The inhibitory effects of RDL on *P. acnes*-induced ear edema in mice were evaluated by measuring the ear thickness (**B**) and ear biopsy weight (**C**). (**D**) The *P. acnes*-injected ear was punched with a 6 mm biopsy punch 24 h after *P. acnes* injection and homogenized in 200 ml of sterile PBS with a tissue grinder. CFUs of *P. acnes* were enumerated by plating serial dilutions of the homogenate on an agar plate. Data represent mean ± SE of four individual experiments. Significant difference from *P. acnes* only treatment (vehicle), * *p* < 0.05 and ** *p* < 0.01.

**Figure 6 ijms-21-01717-f006:** GC-MS Chromatogram of methanol extract of RDL and results from comparison with the library data.

**Figure 7 ijms-21-01717-f007:** Proposed mechanism of *Rosa davurica* Pall. leaves (RDL) extract in the suppression of skin disease. RDL suppresses *P. acnes*-induced inflammatory responses within acne lesions by MAPK and NF-κB signaling. The t-bar denotes an inhibitory effect.

**Table 1 ijms-21-01717-t001:** MICs and MBCs against *P. acnes*, *S. epidermidis* and *S. aureus* for various parts of *Rosa davurica* Pall. and clindamycin. MIC-minimum inhibitory concentration (μg/mL); MBC-minimum bactericidal concentration (μg/mL). Clindamycin was used as a positive control. The experiments were performed in triplicate to determine the MIC/MBC values.

Compound	*P. acnes*	*S. epidermidis*	*S. aureus*
	MIC (μg/ml)	MBC (μg/ml)	MIC (μg/ml)	MBC (μg/ml)	MIC (μg/ml)	MBC (μg/ml)
Leaf	62.5	125	125	125	62.5	250
Root	125	500	125	250	250	500
Stem	500	1000	250	500	250	500
Fruit	1000	1000	500	1000	500	1000
Clindamycin	15.6	31.2	15.6	15.6	15.6	31.2

**Table 2 ijms-21-01717-t002:** The major components and its biological activities obtained through the GC-MS study of RDL. (1) Dr. Duke’s phytochemical and ethnobotanical database.

No.	Retention Time (Min)	Name	Formula	Score (%)	Mass (m/z)	Activity (1)
1	24.393	4H-Pyran-4-one, 2,3-dihydro-3,5-dihydroxy-6-methyl-(=DDMP)	C6H8O4	75.86	144	Antimicrobial anti-inflammatory
2	35.134	1,2,3-Bezenetriol (=pyrogallol)	C6H6O3	94.35	126	Antimicrobial preservative
3	56.038	n-Hexadecanoic acid(=palmitic acid)	C16H32O2	91.33	256.2	Antioxidant, hypocholesterolemic nematicide, pesticide, lubricant, antiandrogenic, flavor, hemolytic 5-alpha reductase inhibitor
4	58.141	1-Hexadecanol(=Cetyl alcohol )	C16H34O	88.25	242.3	Antimicrobial
5	59.142	9,12-Octadecadienoic acid (Z,Z)-(=Linoleic acid)	C18H32O2	95.86	280.2	Antioxidant, Anticancer, Antidiabet, hypocholesterolemic action
6	59.286	9-Octadecadienoic acid, (E)- (=oleic acid)	C18H34O2	90.56	256.2	Antiinflammatory, antiandrogenic cancer preventive, dermatitigenic hypocholesterolemic, 5-alpha reductase inhibitor, anemiagenic insectifuge, flavor
7	59.813	Octadecadienoic acid(=staric acid)	C18H36O2	88.64	284.3	Antidiabet, hypocholesterolemic action

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
