# Peer review of "Rosa davurica Pall. Improves Propionibacterium acnes-Induced Inflammatory Responses in Mouse Ear Edema Model and Suppresses Pro-Inflammatory Chemokine Production via MAPK and NF-κB Pathways in HaCaT Cells"

_ijms, 2020, doi:10.3390/ijms21051717_

Round 1

Reviewer 1 Report

  1. The research and its results are well presented
  2. The authors can elaborate the conclusions little further
  3. Can add few lines in the discussion regarding the scientific significance of the work

Author Response

Thank you very much for your kind reply with warm heart and generosity.

Your suggestions and advices are greatly appreciated and carefully reflected in our revised manuscript as follows.

Comments and Suggestions for Authors:

  1. The research and its results are well presented
  2. The authors can elaborate the conclusions little further
  3. Can add few lines in the discussion regarding the scientific significance of the work

According to the comments, we have added  the scientific significance of our work in the Discussion section and also elaborated the conclusions further in detail for authors and reviewers in the revised manuscript.

We hope our manuscript has been prepared earnestly and sincerely for the publication of this prostigeous journal.

Best regards,

Reviewer 2 Report

No further comment.

Author Response

Thank you very much for your kind reply with warm heart and generosity.

This manuscript is a resubmission of an earlier submission. The following is a list of the peer review reports and author responses from that submission.

Round 1

Reviewer 1 Report

I would recommend the authors to add future insights of this study in the discussion. The authors need to elaborate the discussion further Overall, the manuscript is well written, and methods are well within the scope of manuscript

Author Response

Thank you for your kind and helpful suggestions.

Further works for the effects of various parts of Rose davurica Pall. is needed to study not only anti-acne but also other possible applications to various therapeutic fields.

Reviewer 2 Report

Is "ELAIS" on page 157, line 153 a spelling mistake?

Have you discussed the 72-hours results for Figure 2?

Author Response

Thank you for the valuable comments.

We have corrected the typing error to ‘ELISA’ in the revised manuscript.

In Figure 2, the RDL was processed from 0 to 300 for 24, 48, and 72 hours. RDL did not show any cytotoxicity with the times and concentration tested in the present study.

On the other hand, Significant cell proliferation increase was observed at 48 and 72 hours depending on the concentration. As a result, only the concentrations from 10 to 100 were chosen for the experiment, especially at 24 hours, which had little effect on cell proliferation, and were determined as experimental concentrations.

Reviewer 3 Report

For the in vitro antibacterial assays, there was no positive control(s) (e.g. conventional used antibiotics) included in the study to compared the effects of their tested extracts. Also, MIC or MBC must be included in their study.

For the cytokines inhibitory studies, the authors expressed the results as (% of controls). The absolute values of the controls should be included in their figure legends. Positive controls like steroids should be included in their study and clearly presented in the figures.

For the animal study, positive control should be included for comparison.

Author Response

Thank you for the valuable comments.

In Figure 1, Triclosan has been was included for the experiment as a positive control as suggested by the reviewer. The aims of the experiment was to screen the anti-bacterial activities of various parts of Rose davurica Pall., and the data showed Rose davurica Pall. leaf (RDL) has a potent anti-bacterial activity, which is comparable to triclosan. Hence, we do not importent to do MIC or MBC studies in our manuscript. 

In general, many other studies have also reported papers like as without including positive control.

Ref. [1-3]

[1] Lee, E.H.; Shin, J.H.; Kim, S.S.; Joo, J.H.; Choi, E.; Seo, S.R. Suppression of Propionibacterium Acnes-Induced Skin Inflammation by Laurus Nobilis Extract and its Major Constituent Eucalyptol. Int. J. Mol. Sci. 2019, 20, 10.3390/ijms20143510.

[2] Huang, W.C.; Tsai, T.H.; Chuang, L.T.; Li, Y.Y.; Zouboulis, C.C.; Tsai, P.J. Anti-Bacterial and Anti-Inflammatory Properties of Capric Acid Against Propionibacterium Acnes: A Comparative Study with Lauric Acid. J. Dermatol. Sci. 2014, 73, 232-240.

[3] Hsu, C.; Tsai, T.H.; Li, Y.Y.; Wu, W.H.; Huang, C.J.; Tsai, P.J. Wild Bitter Melon (Momordica Charantia Linn. Var. Abbreviata Ser.) Extract and its Bioactive Components Suppress Propionibacterium Acnes-Induced Inflammation. Food Chem. 2012, 135, 976-984.

Round 2

Reviewer 3 Report

The activities of Rose davurica Pall. on various assays presented by the authors are relative mild. Without positive controls and I think it is hard to give an objective messages to the readers.